# Nitro-Group-Containing Thiopeptide Derivatives as Promising Agents to Target *Clostridioides difficile*

**DOI:** 10.3390/ph15050623

**Published:** 2022-05-19

**Authors:** Dahyun Kim, Young-Rok Kim, Hee-Jong Hwang, Marco A. Ciufolini, Jusuk Lee, Hakyeong Lee, Shyaka Clovis, Sungji Jung, Sang-Hun Oh, Young-Jin Son, Jin-Hwan Kwak

**Affiliations:** 1A&J Science Co., Ltd., 80 Chumbok Ro, Dong Gu, Daegu 41061, Korea; dhk@anjscience.co.kr (D.K.); hwanghj@anjscience.co.kr (H.-J.H.); jslee@anjscience.co.kr (J.L.); hkleee28@gmail.com (H.L.); shyaka@anjscience.co.kr (S.C.); 2School of Life Science, Handong Global University, 558 Handong Ro, Heunghae-Eup, Buk-Gu, Pohang 37554, Korea; fred87@nate.com (Y.-R.K.); sj_411@naver.com (S.J.); shoh@handong.edu (S.-H.O.); 3Department of Chemistry, University of British Columbia, 2036 Main Mall, Vancouver, BC V6K 1Z1, Canada; ciufi@chem.ubc.ca

**Keywords:** *Clostridioides difficile*, thiopeptide antibiotics, nitro-group-containing antibiotics

## Abstract

The US Centers for Disease Control and Prevention (CDC) lists *Clostridioides difficile* as an urgent bacterial threat. Yet, only two drugs, vancomycin and fidaxomicin, are approved by the FDA for the treatment of *C. difficile* infections as of this writing, while the global pipeline of new drugs is sparse at best. Thus, there is a clear and urgent need for new antibiotics against that organism. Herein, we disclose that AJ-024, a nitroimidazole derivative of a 26-membered thiopeptide, is a promising anti-*C. difficile* lead compound. Despite their unique mode of action, thiopeptides remain largely unexploited as anti-infective agents. AJ-024 combines potent in vitro activity against various strains of *C. difficile* with a noteworthy safety profile and desirable pharmacokinetic properties. Its time-kill kinetics against a hypervirulent *C. difficile* ribotype 027 and in vivo (mouse) efficacy compare favorably to vancomycin, and they define AJ-024 as a valuable platform for the development of new anti-*C. difficile* antibiotics.

## 1. Introduction

*Clostridioides difficile* is a Gram-positive, toxin-producing anaerobic bacterium widely distributed in the intestinal tract [1]. *C. difficile* is responsible for serious nosocomial infections that may cause life-threatening complications, such as toxic megacolon, pseudomembranous colitis, and systematic inflammatory response syndrome [2,3,4]. The frequency and severity of *C. difficile* infection (CDI) has been increasing significantly. In the U.S alone, more than 223,900 cases of CDI with 12,800 deaths were reported in 2017 [5]. Moreover, the rapid emergence of the hypervirulent PCR ribotype 027 is associated with significantly higher morbidity and mortality and clinically severe complications [6,7]. *C. difficile* has thus been designated as an urgent threat by the U.S Centers for Disease Control and Prevention (CDC). Therefore, the development of new agents that effectively target *C. difficile* is considered as a top priority [8,9].

Currently, only two antibiotics, vancomycin and fidaxomicin, are approved by the FDA for the treatment of CDI [10]. Vancomycin suffers from dysbiosis, which induces high rates of relapse (25% to 30%). Compared to vancomycin, fidaxomicin has lower recurrence rates because of its better selectivity towards *C. difficile*. However, its high price and poor clinical outcome toward the hypervirulent ribotype 027 remain a challenge. [11,12]. Clearly, the shortage of effective agents to target *C. difficile*, and the increase in treatment failure and recurrent rates call for the development of new agents [13].

In recent times, it has become apparent that effective anti-CDI agents should possess (1) low oral bioavailability, (2) low aqueous solubility, and (3) a new chemical structure and mode of action [14,15], enabling the drugs to selectively target *C. difficile* in the large intestine. In that respect, it is worthwhile to note that metronidazole is no longer recommended for the clinical treatment of CDI, because its high bioavailability leads to undesired systemic toxicity, high treatment failure, and frequent recurrence of diseases. On the other hand, the nitro group in metronidazole is responsible for its mode of action, which involves bioreductive metabolism in bacteria [16,17]. Such a mechanism could still be clinically exploited if an appropriate nitro-group-containing moiety were to be introduced onto a suitable framework already possessing antibiotic activity: the resulting conjugate may well possess superior anti-CDI properties.

In that regard, thiopeptide antibiotics may well be ideal platforms. Thiopeptides are powerfully active against Gram-positive agents. They possess new chemical structures and a clinically unexploited mode of action. For instance, 26-membered thiopeptides bind tightly to a cleft between the bacterial 23S ribosomal RNA subunit and the protein L11, causing inhibition of bacterial protein synthesis [18,19]. Moreover, they are non-toxic to mammalian cells and exhibit very low oral bioavailability [20,21]. Therefore, the introduction of a nitro-containing group in this privileged scaffold may greatly enhance activity against *C. difficile*.

Herein, we describe AJ-024 (Figure 1), a new thiopeptide lead compound that incorporates a nitroimidazole moiety. This agent has been tested against extensively classified *C. difficile* isolates. Cytotoxicity against two mammalian cell lines and preliminary ADME properties have been investigated. Furthermore, time-kill kinetics against a hypervirulent strain *C. difficile* ribotype 027, as well as in vivo efficacy, in a mouse model have been determined.

## 2. Results and Discussion

### 2.1. Chemical Synthesis of AJ-024 and Antibacterial Activity Thereof against Representative C. difficile Strains

Our previous synthetic efforts [22] demonstrated that late-stage Suzuki coupling [23] of macrocycle **6** (Figure 1) with a range of aryl and heteroaryl boronic acids occurs quite efficiently. For instance, coupling of **6** with **4** is the key step for the preparation of AJ-024, and gram quantities of AJ-024 are now available thanks to the chemically convergent sequence of Figure 1.

AJ-024 thus obtained was initially screened with a few representative *C. difficile* strains. Its in vitro activity was found to be better than vancomycin in all the strains tested (Table 1).

### 2.2. Superior In Vitro Activity of AJ-024 against Extensively Classified C. difficile Clinical Isolates

AJ-024 was further screened against *C. difficile* isolates collected in South Korea from 2008 to 2018. These clinical isolates were fully characterized by multilocus sequence typing (MLST), PCR ribotyping and the presence of genes encoding toxin A, toxin B and binary toxin [24,25]. The antibiotic activity of AJ-024 was tested according to MLST clade and ribotype. In vitro drug susceptibility testing was performed by measuring the MIC. AJ-024 was found to be active against all clinical isolates of *C. difficile* and its activity was superior to vancomycin (Table 2). MICs for AJ-024 ranged from 0.25 to 1 μg/mL, with an MIC_50_ of 0.5 μg/mL. Higher MICs were recorded for vancomycin (range 0.5–2 μg/mL) with an MIC_50_ of 1 μg/mL. AJ-024 showed excellent in vitro activity against all of the isolated clinical strains collected in South Korea.

### 2.3. Antimicrobial Activities of AJ-024 against Other Bacterial Species

AJ-024 was further screened against other Gram-positive and Gram-negative pathogens (Table 3). The presence of a nitro-group generally enhanced bacterial activity against Gram-positive pathogens relative to micrococcin P2. However, as with other thiopeptides, antibiotic activity was restricted to Gram-positive pathogens. Noteworthy in that regard is the activity of the compound against vancomycin-resistant *Enterococcus* (VRE). The dual activity of AJ-024 against both organisms may have major clinical implications [26].

### 2.4. AJ-024 Has No Appreciable Human Cellular Toxicity

Cell cytotoxicity experiments were carried out with mammalian cell lines HEK 293T and SH-SY5Y. It is established that thiopeptides are not toxic to mammalian cells. However, the presence of nitro groups might have implications on metabolic and potential toxicity issues. Fortunately, AJ-024 exhibited very low cytotoxicity against the above cells and it was non-toxic up to 50 μg/mL even after 72 h of incubation (Figure 2). The high selectivity index of AJ-024 (antibacterial activity vs. cellular toxicity) is exceptional.

### 2.5. Favorable ADME Properties of AJ-024

A preliminary in vivo ADME comparison of AJ-024 vs. certain marketed drugs that are known to possess metabolic liabilities was carried out to assess potential toxic effects (Table 4). For instance, ketoconazole is a potent CYP3A4 inhibitor, and it is responsible for inducing toxicity through drug–drug interactions (DDI). The metabolic and plasma stability of AJ-024 was evaluated relative to verapamil and procaine. No significant metabolic liabilities were noted. Incubation of AJ-024 for 30 min with human liver microsomes and with human plasma proteins returned 63.8% and 95.9% of unchanged compound, respectively. This indicates that AJ-024 possesses high metabolic and plasma stability in humans. Moreover, potential drug–drug-interactions (DDI) are expected to be minimal, as the compound exhibits relatively high stability toward the various cytochrome P450 enzymes that are responsible for drug metabolisms.

### 2.6. Time-Kill Assay of AJ-024

A comparison of the killing kinetics of AJ-024 and vancomycin was carried out with a hypervirulent strain of *C. difficile* ribotype 027. The killing effect for AJ-024 and vancomycin is shown in Figure 2, together with reductions in CFU/mL for *C. difficile* ribotype following a 24 h exposure to the antibiotics. The starting inoculum for *C. difficile* was Log_10_ 5.62 CFU/mL. It is apparent that AJ-024 kills *C. difficile* ribotype 027 more efficiently than vancomycin. AJ-024 reached the LOD and showed a bactericidal activity, defined as ≥3 log_10_ CFU/mL reduction in viability relative to the starting inoculum, at concentrations of 1×, 2× and 4× MIC. At 4× MIC, bactericidal activity was maintained for up to 24 h. At 1× and 2× MIC, regrowth occurred at log_10_ 2.88 CFU/mL and log_10_ 2.62 CFU/mL at 24 h, respectively. However, at 2× MIC, bactericidal activity was maintained, with ≥3 log_10_ CFU/mL reduction. On the other hand, vancomycin is only bacteriostatic at concentration of 1×, 2× and 4× MIC (Figure 3).

### 2.7. Effect of AJ-024 on Acute CDI In Vivo Infection Model

The favorable activity, toxicity and metabolic profiles of AJ-024 prompted us to investigate its in vivo efficacy vs. the FDA approved drug, vancomycin in a mouse model. We adopted the established model of Zhou et al. [27], in which mice are initially treated with dextran sulfate sodium (DSS) to induce IBD, then infected with *C. difficile* to induce CDI—inflammatory bowel disease (IBD) comorbidity. The severity of the ensuing infection is apparent from the fact that all untreated mice succumbed within 6 days. Since clinical cases of CDI patients suffering from IBD are on the rise [28,29,30], this model should be a good indicator of the therapeutic potential of AJ-024 for the treatment of CDI-IBD.

AJ-024 and vancomycin were administrated once daily at a dose of 30 mg/kg via oral gavage. Both resulted in a survival rate of 80% over the observed period of 15 days. This clearly suggests that AJ-024 holds significant potential as a treatment for *C. difficile* infection (Figure 4).

## 3. Materials and Methods

### 3.1. Chemistry

All commercial reagents (Acros Organics, Morris Plains, N.J., USA; Sig-ma-Aldrich, St Louis, MI, USA; Alfa aesar, Ward Hill, MA, USA; TCI, Tokyo, Japan) were used as received. Thin-layer chromatography (TLC) was carried out with Macherey-Nagel precoated plates (POLYGRAM^®^SIL/UV254). Flash chromatography employed Merck silica gel 60 (40–63 μm) and technical grade solvents. NMR spectra were recorded on Bruker AV VIII 400 spectrometers in the solvents indicated. High-resolution mass spectra (HRMS) were recorded on an AB SCIEX Q-TOF 5600 mass spectrometer. Medium pressure liquid chromatography (MPLC) purifications are performed by Com-biFlash RF+ UV (Redisep^®^ from Teledyne Isco, Lincoln, NE, USA). Purity of the AJ-024 has been determined by HPLC on Kinetex 5 μm biphenyl, 100 Å (GX-281 HPLC system, Gilson, Middleton, WI, USA; column tube 250 mm × 21.2 mm i.d.), with ACN/H2O (0.1% TFA) as eluents. The full characterizations of compounds have been made and are available in the Appendix A.

#### Synthesis of AJ-024

##### Synthesis of Methyl (2-bromothiazole-4-carbonyl)-L-threoninate (Compound **2**)

To 2-bromothiazole-4-carboxylic acid (compound **1**, 7.7 mmol, 1.6 g) in CH_2_Cl_2_ (70 mL) was added L-threonine methyl ester hydrochloride (9.2 mmol, 1.56 g, 1.2 equiv.), triethylamine (30.8 mmol, 4.3 mL, 4 equiv.), HOBt (11.6 mmol, 1.56 g, 1.5 equiv.), EDCI (11.6 mmol, 2.2 g, 1.5 equiv.). The mixture was stirred at room temperature for overnight. The reaction was quenched by the addition of water, and the aqueous layer was extracted with CH_2_Cl_2_. The combined organic phase was dried Na_2_SO_4_ and concentrated under vacuum. The crude material was purified by MPLC (0% → 1% MeOH: 100% → 99% CH_2_Cl_2_) to afford the desired compound **2** as a white solid (6.2 mmol, 2 g, 80%), **^1^H NMR** (400 MHz, CDCl_3_) δ 8.11 (s, 1H), 7.89 (d, J = 8.9 Hz, 1H), 4.77 (dd, J = 9.2, 2.5 Hz, 1H), 4.54–4.44 (m, 1H), 3.81 (s, 3H), 2.51 (d, J = 5.2 Hz, 1H), 1.30 (d, J = 6.4 Hz, 3H); **^13^C NMR** (101 MHz, CDCl_3_) δ 171.0, 160.2, 149.2, 136.1, 127.7, 68.0, 57.4, 52.8, 20.0; **HRMS (FAB+)** Calcd for C_9_H_12_BrN_2_O_4_S^+^ [M+H]^+^ 322.9696, found 322.9699.

##### Synthesis of 2-bromo-N-((2S,3R)-3-hydroxy-1-((2-(2-methyl-5-nitro-1H-imidazol-1-yl)ethyl)amino)-1-oxobutan-2-yl)thiazole-4-carboxamide (Compound **3**)

To compound **2** (6.2 mmol, 2 g) in THF (30 mL) and water (30 mL) was added lithium hydroxide monohydrate (15.5 mmol, 767 mg, 2.5 equiv.). The mixture was stirred at room temperature for 4 h. Upon complete conversion, the reaction mixture was acidified with 1M HCl and extracted with ethyl acetate. The combined extracts were dried over Na_2_SO_4_ and evaporated. The resulting crude product was immediately treated with 2-methyl-5-nitro-1H-imidazole-1-ethylamine (7.44 mmol, 1.54 g, 1.2 equiv.) and EDCI (9.3 mmol, 1.78 g, 1.5 equiv.), HOBt (9.3 mmol, 1.26 g, 1.5 equiv.), triethylamine (25 mmol, 3.5 mL, 4 equiv.) in CH_2_Cl_2_ (75 mL), and the mixture was stirred at room temperature overnight. Purification of the crude product by MPLC (2% → 4% MeOH: 98% → 96% CH_2_Cl_2_) furnished the desired compound **3** as a white solid (2.1 mmol, 968 mg, 34%); **^1^H NMR** (400 MHz, DMSO-d_6_) δ 8.36 (s, 1H), 8.25 (t, J = 5.9 Hz, 1H), 8.01 (s, 1H), 7.85 (d, J = 8.8 Hz, 1H), 5.17 (d, J = 5.2 Hz, 1H), 4.37–4.24 (m, 2H), 4.20 (dd, J = 8.8, 3.1 Hz, 1H), 4.10 (m, 1H), 3.52–3.43 (m, 2H), 2.38 (s, 3H), 1.02 (d, J = 6.3 Hz, 3H); **^13^C NMR** (101 MHz, DMSO-d_6_) δ170.8, 159.6, 152.1, 149.4, 138.9, 137.1, 133.7, 129.6, 66.6, 58.7, 45.8, 38.7, 20.9, 14.3; **HRMS (****EI+)** Calcd for C_14_H_17_BrN_6_O_5_S^+^ [M+H]^+^ 461.0237, found 461.0189.

##### Synthesis of (Z)-2-bromo-N-(1-((2-(2-methyl-5-nitro-1H-imidazol-1-yl)ethyl)amino)-1-oxobut-2-en-2-yl)thiazole-4-carboxamide (Compound **4**)

To compound **3** (1.7 mmol, 790 mg) in CH_2_Cl_2_ (20 mL) was added methanesulfonyl chloride (5.1 mmol, 0.4 mL, 3 equiv.) and triethylamine (6.84 mmol, 0.96 mL, 4 equiv.). The mixture was stirred at room temperature for 1 h, then DBU (34.2 mmol, 5.1 mL, 20 equiv.) was added and the mixture was further stirred at room temperature overnight. The reaction was quenched by the addition of sat.aq NH_4_Cl and extracted with CH_2_Cl_2_. The organic layer was dried over Na_2_SO_4_ and evaporated, and the crude product was purified by MPLC (2% → 3% MeOH: 98% → 97% CH_2_Cl_2_) to afford the desired compound **4** as a white solid (1.05 mmol, 465 mg, 50%),**^1^H NMR** (400 MHz, CDCl_3_) δ 8.37 (s, 1H), 8.14 (s, 1H), 7.93 (s, 1H), 6.77–6.67 (m, 2H), 4.50 (t, J = 6.5 Hz, 2H), 3.67 (q, J = 6.3 Hz, 2H), 2.54 (s, 3H), 1.83 (d, J = 7.1 Hz, 3H); **^13^C NMR** (101 MHz, CDCl_3_) δ 165.3, 158.4, 151.4, 148.7, 138.4, 136.5, 133.5, 131.8, 128.8, 128.4, 44.8, 39.6, 14.1, 13.8; **HRMS (EI+)** Calcd for C_14_H_16_BrN_6_O_4_S^+^ [M+H]^+^ 443.0132, found 443.0083.

##### Synthesis of ((12Z,32Z,72Z,112Z,4S,8S,12Z,15S)-26-(4-bromothiazol-2-yl)-12-ethylidene-4,15-bis((R)-1-hydroxyethyl)-8-isopropyl-5,9,13,16-tetraaza-1(2,4),3,7,11(4,2) -tetrathiazola-2(3,2)-pyridinacycloheptadecaphane-6,10,14,17-tetraone (AJ-024)

To compound **5** (prepared as described by Hwang et al. [22]; 1 mmol, 966 mg) in dioxane (10 mL) was added B_2_Pin_2_ (1.3 mmol, 330 mg, 1.3 equiv.), potassium acetate (1.5 mmol, 147 mg, 1.5 equiv.), and XPhos (0.15 mmol, 71 mg, 0.15 equiv.). The solution was purged with N_2_ for few minutes, then Pd_2_(dba)_3_ (0.1 mmol, 91 mg, 0.10 equiv.) was added and the mixture was stirred at 90 °C for 1.5 h. The mixture was cooled to room temperature, diluted with CH_2_Cl_2_ (30 mL) and washed with H_2_O (10 mL). The organic layer was collected, filtered through celite, dried over Na_2_SO_4_, and concentrated in vacuo. The resulting crude boronic acid **6** was immediately dissolved in a mixture of THF (8 mL)/water (2 mL) and treated with compound **4** (1.1 equiv.) and K_2_CO_3_ (3 mmol, 412 mg, 3.0 equiv.). The solution was purged with N_2_ for few minutes, then Pd(dtbpf)Cl_2_ (0.01 mmol, 64 mg, 10 mol %) was added and the mixture was warmed to 40 °C and stirred overnight. The mixture was filtered through Celite, and the Celite bed was further washed with ethyl acetate. The combined filtrates were washed with 1M HCl, dried over Na_2_SO_4_, and concentrated in vacuo. The crude product was purified by MPLC (2% → 5% MeOH: 98% → 95% CH_2_Cl_2_) to afford **AJ-024** as a yellow solid (656 mg, 53% Yield), **^1^H NMR** (400 MHz, CDCl_3_) δ 9.08 (s, 1H), 8.64 (d, J = 9.4 Hz, 1H), 8.42 (s, 1H), 8.28 (s, 1H), 8.18–8.11 (m, 2H), 8.09–8.02 (m, 2H), 7.98–7.85 (m, 4H), 7.78 (s, 1H), 7.71 (d, J = 7.9 Hz, 2H), 6.76 (q, J = 7.0 Hz, 1H), 6.39 (q, J = 6.9 Hz, 1H), 5.26–5.09 (m, 2H), 4.69 (d, J = 6.4 Hz, 1H), 4.62–4.57 (m, 2H), 4.33 (d, J = 6.1 Hz, 2H), 3.97–3.85 (m, 1H), 3.76–3.58 (m, 1H), 2.55 (s, 3H), 2.10–1.95 (m, 1H), 1.85 (d, J = 7.0 Hz, 3H), 1.80 (d, J = 7.1 Hz, 3H), 1.45 (d, J = 6.3 Hz, 3H), 1.21–1.06 (m, 6H), 0.93 (d, J = 6.6 Hz, 3H);^**13**^**C NMR** (101 MHz, CDCl_3_) δ 170.7, 170.1, 168.2, 168.0, 165.9, 165.8, 165.4, 162.9, 161.2, 160.6, 160.4, 159.6, 153.5, 151.5, 150.9, 150.3, 149.8, 149.5, 149.4, 149.1, 148.7, 140.1, 138.6, 132.8, 130.9, 129.7, 129.0, 127.5, 127.0, 124.9, 124.2, 123.7, 121.7, 121.6, 118.2, 68.0, 67.2, 58.0, 55.8, 55.4, 45.5, 40.0, 33.5, 29.7, 19.8, 19.4, 19.0, 18.1, 14.9, 14.1, 13.5; **HRMS (FAB+)** Calcd for C_51_H_51_N_16_O_10_S_6_^+^ [M+H]^+^ 1239.2293, found 1239.2294.

**HPLC****:** Analyzing condition: Flow rate 1.0 mL / min, A: 95% to 0% Water (0.1% TFA), B: 5% to 100% ACN (0.1% TFA) linear gradient in 10 min − retention time = 6.5 min, purity 99%.

Minimum inhibitory concentrations (MICs) were determined by a two-fold agar dilution method as described by the Clinical and Laboratory Standards Institute (CLSI) [31,32]. Vancomycin (Sigma-Aldrich, St. Louis, MI, USA) was diluted in distilled water and AJ-024 was diluted in DMSO. Doubling dilutions of vancomycin and AJ-024 (0.06~8 μg/mL) were made in Brucella agar supplemented with 5% defibrinated sheep blood, 1 μg/mL hemin and 5 μg/mL vitamin K1. Test organisms were grown on Brain Heart Infusion broth (BHI, Difco™) plates containing 5% defibrinated sheep blood and sub-cultured into Brain Heart Infusion broth (BHI, Difco™) with 0.5% yeast extract (Difco™) and incubated for 24 h at 37 °C in an anaerobic chamber. The cultured bacteria were diluted using the same fresh medium to achieve a bacterial cell density of approximately 10^7^ colony-forming units (CFU)/mL. All test organisms were seeded in a supplemented Brucella agar at a density of 10^5^ CFU/spot. The plates were incubated at 35 °C for 48 h and were examined for bacterial growth.

### 3.2. Cellular Toxicity

Cell cytotoxicity was evaluated against the human cell lines SH-SH5Y, HEK293T using the CCK-8 cell viability assay. Each cell was grown in Dulbecco’s Modified Eagles Medium (DMEM; Gibco, New York, NY, USA) supplemented with 1% penicillin/streptomycin and 10% fetal bovine serum (Gibco) at humidified incubator adjusted 5% CO_2_ at 37 °C. For cellular toxicity test, SH-SY5Y and HEK 293T cells were plated at a density of 7000 cells/mL in 100 µL of cell culture medium in a 96-well microplate. AJ-024 were treated in triplicate with 5 μM to 40 μM to the cells for 48 h at 37 ℃. Finally, 10 μL of the CCK-8 reagent (Dojindo, Rockville, MD, USA) was added into each well, and Viable cell signal was measured at 450 nm using a microplate reader (Biotek, Vermont, VT, USA). The cell viability was normalized to DMSO control.

### 3.3. Time-Kill Assays

The time-kill studies were performed by the CLSI method [32]. Test organisms were incubated in BHI broth containing 0.5% yeast extract and 0.025% (*w*/*v*) L-cysteine (sigma-aldrich) for 18 h at 37 ℃ in an anaerobic chamber, and they were diluted with fresh broth to approximately 10^5^ CFU/mL. Vancomycin was diluted in distilled water and AJ-024 was diluted in 1% (*v*/*v*) DMSO. Vancomycin and AJ-024 were added to the cultures of 1/2, 1, 2, and 4x the MIC. Samples were removed at 0, 3, 6, and 24 h post inoculation, and serial 10-fold dilutions were performed. The number of viable cells were determined on BHI agar containing 0.5% yeast extract and 5% defibrinated sheep blood after 24 h of incubation. The plates were incubated at 37 ℃ in an anaerobic chamber. Antimicrobials were considered bactericidal at the lowest concentration that reduced the original inoculum by 3 log 10 CFU/mL at each of the time periods and bacteriostatic if the inoculum was reduced by 0 to 3 log 10 CFU/mL.

### 3.4. In Vivo Bacteria Inoculation

*C. difficile* ribotype 027 was grown on blood agar plate and incubated for two days at 37 ℃ in an anaerobic chamber and sub-cultured on 70:30 sporulation medium to facilitate the sporulation of C. difficile ribotype 027. Briefly, the sporulation medium consists of 70% SMC (Bacto peptone 90 g, protease peptone 5 g, NH4SO4 1 g, tris base 1.5 g, agar 15 g per liter) and 30% BHIS (BHI agar with 0.5% yeast extract) agar. For inoculation, bacterial colonies were suspended in 0.9% saline solution and further diluted with saline solutions to adjust OD (Optical Density) 1.0 at 1/10 of original saline solutions. Groups of five male C57BL/6J mice were injected orally with a single 0.5 mL dose of the bacterial suspension.

### 3.5. Mouse IBD-C. difficile Comorbidity Infection Model

All animal experiments were conducted in accordance with the ethical guidelines of the Ethics Review Committee for Animal Experimentation at Handong Global University (Republic of Korea, protocol #HGU-20211214-20). C57BL/6J male (4-week-old male weighing 20 to 22 g, Daehan Bio Link Co., Ltd., Eumseong, Korea) 5 mouse per group was used in C. difficile infection model. Mice were maintained in animal chambers kept at 23 ± 2 ℃ with 55% ± 20% relative humidity. An amount of 2% dextran sodium sulfate (DSS, MP biomedical, Irvine, CA, USA) was inoculated by drinking water for 5 days to cause colitis in mice. A single dose of clindamycin (20 mg/kg) was injected intraperitonially a day before C. difficile ribotype 027 challenge. An amount of 30 mg/kg of each drug (vancomycin, AJ-024) were dissolved in a specified vehicle and 0.25 mL was inoculated after 6 h of infection through oral gavage, and in every 24 h for 6 days.

### 3.6. Inhibition of CYP Enzymes Activity by AJ-024

All incubations were performed in duplicate, and the mean values were used for analysis. The activity of five different substrates, phenacetin 50 μM, diclofenac 10 μM, S-mephenytoin 100 μM, dextromethorphan 5 μM, midazolam 2.5 μM were determined as probe activities for CYP1A2, CYP2C9, CYP2C19, CYP2D6 and CYP3A, respectively, in 0.1 M phosphate buffer (pH 7.4). Then, AJ-024 at different concentrations (0, 10 uM) was added and pre-incubated for 5 min before an NADPH regenerating system was added. The latter was further incubated at 37 °C for 15 min. The reaction was quenched by placing the incubation tubes on ice and adding 40 µL of ice-cold acetonitrile. The mixtures were then centrifuged at 15,000× *g* for 5 min at 4 °C. Aliquots of the supernatant were injected onto an LC-MS/MS system. The CYP-mediated activities in the presence of a known CYP3A4 inhibitor, ketoconazole, was expressed as percentages of the corresponding control values. Metabolites of each CYP coenzyme indicator drug generated through the above reaction were analyzed using the Shimadzu Nexera XR system and TSQ vantage (Thermo). The HPLC column was a Kinetex C18 column (2.1 × 100 mm^2^, 2.6 μm particle size; Phenomenex, Torrance, CA, USA). The generated metabolites were quantified using MRM (Multiple Reaction Monitoring) quantification modes, and data were analyzed with Xcalibur version 1.6.1 (Thermo Fisher Scientific Inc., Waltham, MA, USA).

### 3.7. In Vitro Metabolic Stability of AJ-024

The metabolic stability of AJ-024 and verapamil (rapid clearance control) were measured using human liver microsomes. Microsome incubation was performed in the final incubation mixture, 0.5 mg/mL human liver microsome, 0.1 M potassium phosphate buffer (pH 7.4), and NADPH regeneration system (1 mM NADPH, 10 mM MgCl_2_). In the pre-incubation, AJ-024 or Verapamil (positive control), HLM were mixed and incubated in a Thermomixer (Eppendorf, Hamburg, Germany) under 350 rpm, 37 ℃, 5 min. Reactions were initiated by the addition of 1 mM NADPH and then quenched by the addition of 40 μL of ice-cold acetonitrile containing 10 μM chlorpropamide (CPP) as an internal standard at 0 and 30 min. After quenching, centrifugation of the incubation mixture was performed at 15,000× *g* for 5 min at 4 ℃. An aliquot of 2 μL of the supernatant was injected into the LC-MS/MS system and analyzed in Xcalibur version 1.1.1 (Thermo Fisher Scientific Inc., Waltham, MA, USA).

### 3.8. Plasma Protein Binding Study of AJ-024

The protein binding of human plasma of AJ-024, dexamethasone (low PPB control), and warfarin (high PPB control) was evaluated by equilibrium dialysis technique using rapid equilibrium dialysis (RED) device (Thermo Fisher Scientific Inc., Waltham, MA, USA). Before use, the Teflon base plates were carefully washed with soap and water and then soaked in 70% ethanol overnight to minimize bacterial growth. Each compound was spiked into blank human plasma to yield a final concentration of 10 μM. Then, 200 μL of the spiked plasma samples was placed into the sample chamber, and a 350 μL aliquot of isotonic phosphate-buffered saline was placed into the adjacent chamber. Each plate was tightly sealed with a self-adhesive lid and sealing tape to prevent evaporation, and then incubated at 37 ℃ with shaking at a rate of 350 rpm. After 4 h incubation, samples were analyzed by LC–MS/MS. The fraction unbound in plasma (fp) was calculated by dividing the target concentrations in ‘the buffer’ compartment (Cf) by those of the plasma compartment (Cp).

## 4. Conclusions

*C. difficile* is designated as an urgent threat by the U.S CDC; yet, only two drugs, vancomycin and fidaxomicin, are available to treat it. Because of the foregoing, there is a clear and urgent need for new, effective antibiotics against *C. difficile*. We have identified AJ-024, a 26-membered thiopeptide derivative with new chemical structure and mode of action, as a promising lead compound for CDI. The introduction of nitro-group has been successfully leveraged to enhance activity against extensively classified *C. difficile* clinical isolates collected in S. Korea. Moreover, the superior in vitro activities of AJ-024 to vancomycin has been observed while minimizing potential metabolic and toxicity liabilities. Its effective time-kill kinetics through bactericidal effect, along with comparable in vivo efficacy to vancomycin CDI-IBD comorbidity model suggest its therapeutic potential for the treatment of CDI-IBD. All these promising results clearly warrant further clinical development of AJ-024 as an effective agent to target *C. difficile*. Results of SAR and optimization studies of AJ-024 and its analogs will be reported in due course.

## Data Availability

Not Applicable.

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
