# Peer review of "Nitro-Group-Containing Thiopeptide Derivatives as Promising Agents to Target Clostridioides difficile"

_pharmaceuticals, 2022, doi:10.3390/ph15050623_

Round 1

Reviewer 1 Report

Dear Authors,

It was pleasure to read your manuscript devoted to the thiopeptide derivatives, active against C.difficile and other gram-positive bacteria, including vancomycin resistant strains. These  derivatives can be used for further development of the novel antibacterials - one of the most urgent tasks in the era of increasing drug-resistance of bacteria.  

There are several questions that i would like to be clarified:

  1. Figure 2 - i would like to see comparison of the toxicity of the AJ-024 to the clinically used antibiotics, including VAN
  2. Table 4 - will it be possible to provide comparison with VAN
  3. Could you please clarify why the antibacterial stress was made on C. difficile? It seems that the antibacterial activity of AJ-024 is broad.

Thank you for the comments

I hope that the manuscript will be accepted after the comments will be provided.

Sincerely yours,

Referee

Author Response

  1. Cellular toxicity studies on vancomycin have been reported in the literature, and its IC50 value is determined to be at 1,815 μM in the HEK293T cell line.1 Generally, vancomycin and many other antibiotics in the clinics are known to be cellularly non-toxic. Since AJ-024 is an investigational agent, cellular toxicity studies must be performed.
  2. We respectfully decline to do so because the point of this ADME experiment is to verify whether AJ-024, which has a metabolically labile nitro group, is a pharmacologically stable compound. Therefore, the authors provided the comparison data between the marketed drugs that are known to possess certain metabolic liabilities. For instance, ketoconazole is a potent CYP3A4 inhibitor. Verapamil has a low bioavailability due to its low hepatic stability that causes rapid metabolic elimination. The similar logic applies to procaine that has a very low plasma stability. Therefore, one could conclude that AJ-024 possesses relatively safe ADME profiles
  3. Extensive studies on AJ-024 is in progress, but the authors have noted that AJ-024 possesses suitable pharmacokinetic properties to specifically target difficile residing in large intestines. Majority of the global C. difficile investigational agents in clinical developments possess low solubility, and low bioavailability, and AJ-024 have similar pharmacological profiles to those in the clinical development. In this light, AJ-024 exhibited a comparable in vivo data to that of vancomycin.

Reviewer 2 Report

Reviewer comments

This manuscript describes “Nitro-Group Containing Thiopeptide Derivatives as Promising Agents to Target Clostridioides difficile”.  This is an interesting work and well written article for targeting Clostridioides difficile using nitroimidazole derivative of a 26-membered thiopeptide. AJ-024 seems an interesting lead compound in this regard. However, there are some minor and major issues in the current manuscript. This article can be considered for publication after addressing following concerns.  

Major and Minor concerns:

  • In vitro metabolic stability of AJ-024 profiling, authors need to comment about its cleavage/metabolic sites if metabolite(s) was observed.
  • Also, metabolic stability needs to be discussed in detail in results and discussion based on obtained data. Current discussion is very vague.
  • Analytical purity chromatogram of the compound AJ-024 should be provided in SI file.
  • Conclusions need to be more informative so readers can understand this work.
  • All reference should be in uniform pattern.

Author Response

  1. As delineated in the section 3.8, LC-MS/MS system on this particular in vitro metabolic stability tests are designed to measure only the parent molecule. Therefore, in depth metabolite analysis could not be addressed at this point.
  2. More discussion has been added in manuscript.
  3. The SI now includes an HPLC trace as well as a purity assessment of AJ-024.
  4. More discussion has been added in the conclusion section.
  5. Reference format has been uniformly addressed.

References

1 Becerir, T.; Tokgün, O.; Yuksel, S. The Therapeutic Effect of Cilastatin on Drug-Induced Nephrotoxicity: A New Perspective. Eur. Rev. Med. Pharmacol. 2021, 25, 5436-5447.
